# Pollution and Release Characteristics of Nitrogen, Phosphorus and Organic Carbon in Pond Sediments in a Typical Polder Area of the Lake Taihu Basin

Changkang Peng [1], Ya Gao [1], Yaqin Tan [2], Genming Sheng [2], Yang Yang [1], Jiong Huang [1], Dayong Sun [2], Daofang Zhang [1,*], Hong Tao [1] and Feipeng Li [1,*]

[1] School of Environment and Architecture, University of Shanghai for Science and Technology, Shanghai 200093, China; pck9125@163.com (C.P.); gaoya_sc@163.com (Y.G.); 193821900@st.usst.edu.cn (Y.Y.); huangjiong0304@163.com (J.H.); taohong@usst.edu.cn (H.T.)

[2] Shanghai Investigation, Design and Research Institute Co., Ltd., Shanghai 200439, China; tyq@sidri.com (Y.T.); sgmlf@126.com (G.S.); sdy@sidri.com (D.S.)

* Correspondence: zhangdf-usst@163.com (D.Z.); lifeipeng@usst.edu.cn (F.L.)

**Abstract:** There is currently a lack of knowledge on the release characteristics of nutrients from artificial pond sediments in polder areas, resulting in problems in future management of such environments, including converting polders to lakes. In this study, sediment samples were taken from a fish pond and a lotus pond in a typical polder area of the Lake Taihu Basin in China. The total nitrogen (TN, 1760–1810 mg/kg), total phosphorus (TP, 1370–1463 mg/kg) and total organic carbon (TOC, 10.1–21.2 g/kg) contents were significantly higher than those found in sediments from the adjacent aquatic system, which indicates that the legacy of agricultural activities has had an obvious cumulative effect on pond sediment nutrients. The release behavior of TN, TP and TOC varied significantly, not only under disturbed and static conditions, but also from sediments sampled at different ponds and depths. During the disturbing condition, there were continuous releases of carbon and nutrients in the lotus pond sediments, while the fish pond sediments showed a higher release at the beginning. Under static release conditions, the release of TP in the surface and bottom sediments of the fish pond increased first, then decreased and stabilized within 24 h, while the release of the lotus pond showed a slow upward trend. Despite the lower concentration of nutrients and TOC, the lotus pond sediment showed a higher release rate. The results suggested that it is necessary to adopt different strategies for different types of ponds in the project of returning polders to lakes; it is especially important to pay attention to the release of nutrients from the bottom sediments of lotus ponds in the project management.

**Keywords:** pond; sediment; nutrient; total organic carbon; polder area

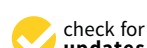

## 1. Introduction

As a main geographical unit at the lowland areas of large aquatic ecosystems, polder is widely known for its advantage in mitigating flood risk by reducing water levels [1,2]. Agricultural activities, through the conversion of natural lake areas to aquacultural production and aquatic cash crop planting, have become one of the important factors for the development and utilization of polder areas [3–5]. Within these intensive agricultural activities (fertilization and irrigation), nutrients from ponds in polder systems may contribute to the eutrophication and harmful algal blooms in surrounding rivers and lakes through excessive amounts of sediment and nutrients in runoff water [6–9]. The estimated results from 2539 polders during 2013 in Lake Taihu Basin showed an annual polder phosphorus export of 1916.2 t and nitrogen export of 16,296 t [3,10]. The eutrophication due to excess nutrient loading is still a worldwide challenge, especially in the polders [11,12], such as the Yangtze and Mekong rivers in Asia, the Rhine and Danube rivers in Europe and Mississippi River in North America [10].

The sediment of ponds is known to be nutrient reservoirs with a 50- to 500-fold rise compared to those in the overlying water [13]. The accumulated nutrients, caused by the settling of particulate material of agricultural activities in the ponds, imparts consistent nutrient loading to the overlying water column and surrounding water environment even after external loading has been reduced [14,15]. In several case studies, the practices of returning polder to lake showed a positive role in enhancing regulation and storage kinetic energy and improving the water ecological environment [16]. Removing layers of surface sediments was a common practice in the project of returning polder to lake to mitigate years of decayed plant and animal matter [17]. There is ample evidence in the literature [18–20] that dredging and dredged areas can play significant roles in the aquatic habitat such as causing increased water turbidity and secondary pollution of Lake Taihu. Moreover, the nutrients generated by remineralization of organic matter within the sediments can be exchanged with the overlying water by a number of mechanisms [21,22]. To mitigate any negative impact on eutrophication of the surrounding aquatic system, it is therefore vitally important to understand the release characteristics of internal nutrients between dredged and undredged sediment to suppress the risk of internal release in a project of returning polder to lake.

Many studies have found that the nutrient release capacity is closely related to the physical and chemical characteristics of the sediments [14,23]. A few studies have been carried out concerning the nutrient release characteristics of the sediments, especially the kinetic characteristics [22]. The ammonium adsorption–desorption equilibrium concentration has been considered as the ammonium release threshold to assess the release risk in the sediments [24]. Although the amount of nutrients released from sediments has been estimated in several instances of literature, the majority of them are focused on results of natural lakes and rivers [25] or artificial lakes [14]. Most of them focused on the physical and chemical properties of sediments and their effects on the nutrient release capacity, release rate and environmental influencing factors of nutrient release from sediments [25–27]. However, the characteristics of nutrients released from pond sediments are obviously different from those of natural lakes without intensive agricultural activity. In view of the agricultural ponds, numerous studies have focused on nitrogen and phosphorus balance [28], nutrient dynamics in water and sediment [29], sediment microbial communities [30], pollution characteristics and evaluation in sediments [31]. There are few studies on nutrient release from pond sediments. The study in [32] examined the effects of flow velocity on nutrient release at the aquaculture pond sediment water. However, there is a lack of investigation of the release characteristics of nutrients from pond sediments.

The project of returning polder to lake was proposed by the Chinese government as specific programs in order to achieve best management practices in minimizing nutrient loading and ensuring the ecological security of Lake Taihu. Therefore, in this study, we took Zhushanwei polder in Lake Taihu, which was one of the reclamation areas, as a case study. The contents of total nitrogen (TN), total phosphorus (TP) and total organic carbon (TOC) in two typical agricultural ponds were investigated. In order to characterize the potential of nutrient release from sediment in different ponds in Zhushanwei polder, the release characteristics of carbon and nutrients were studied in disturbance and static release experiments, respectively. The equations expressing the relationship between carbon and nutrient amount and release time were built.

## 2. Materials and Methods

### 2.1. Study Area and Sample Collection

Zhushanwei polder is one of the reclamation areas located in the west of Zhushan Bay in Lake Taihu. It was built in the early 1970s, covering an area of about 4.2 km$^2$. In order to restore the flood regulation and storage capacity and water environment improvement of Zhushan Bay, the project of returning Zhushanwei polder to Lake Taihu has passed the planning and design approval stage. However, years of fish farming and lotus root

planting activities in Zhushan Bay in Taihu Lake have led to serious organic pollution in sediments of polder ponds [15].

Based on the distribution of ponds in the study area, two types of ponds were selected, a fish pond and a lotus pond (*Neumbo nucifera*, Gaertn). The samples were collected with a column sampler (XDB0204, Beijing New Landmark Soil Equipment Co., Ltd., Beijing, China) with a sampling depth of 0.5 m based on the silting condition. After the overlying water was removed with a siphon, the sediment samples were divided into three parts with sampling depth of 0–10 cm, 10–30 cm and 30–50 cm, respectively, from the surface to the bottom layer. The original samples were recorded and then the following were removed: the parts in contact with the sampler, stones, animals and plants, etc. Being sealed into the plastic Ziplocs, the sediment samples were stored and transported below 4 °C. The sediment samples were freeze-dried and stored for future use after passing through a 100-mesh screen.

*2.2. Experimental Procedure*

The concentrations of total organic carbon (TOC), dissolved total nitrogen (DTN), dissolved total phosphorus (DTP) and ammonia nitrogen ($NH_3$-N) in the overlying water were measured during the release experiments under disturbed and static conditions. The protocols of both disturbed and static release experiments are described below.

2.2.1. Release in Sediment under Disturbed Conditions

The dried and powdered sediments (0.5 g) from each sample and ultrapure water (50 mL) were poured into a 100 mL centrifuge tube which was closed and then shaken on a constant temperature shaker at $25 \pm 1$ °C. Water samples were taken at 0, 5, 30, 60, 90, 120, 180 and 300 min, respectively, and then centrifuged at 5000 rpm for 15 min. The supernatant was passed through 0.45 μm fiber filter membrane and then detected within 24 h.

After measuring the concentrations of TOC, DTN, DTP and $NH_3$-N, the release amount under disturbed conditions was calculated using the following Formula (1) [23]:

$$R_d = \frac{1000(C_e - C_0)V_d}{w_d} \tag{1}$$

where $R_d$ denotes the amount of nutrients (TOC, DTN, DTP and $NH_3$-N, mg/kg) released from pond sediment under disturbed conditions; $C_e$ denotes the nutrition concentration (mg/L) in the solution and $C_0$ denotes the initial concentrations in the solution; $V_d$ denotes the volume (50 mL) of the solution and $W_d$ denotes the quality of the dried sediment (mg) in the release experiment under disturbed conditions.

2.2.2. Release in Sediment under Static Conditions

The dried and powdered sediments (50 g) from each sample and ultrapure water (800 mL) were poured into a 1000 mL beaker sealed with plastic wrap to prevent evaporation. Upper water (150 mL) was taken from the beaker at 0, 5, 30, 60, 90, 120, 180 and 300 min, respectively, and pure water was added to maintain 800 mL in the beaker. The water samples were poured into a 100 mL centrifuge tube centrifuged at 5000 rpm for 15 min. The supernatant was filtered through 0.45 μm fiber filter membrane and tested within 24 h.

After measuring the concentrations of TOC, DTN, DTP and $NH_3$-N, the sediment release under static conditions was calculated using the following Formula (2):

$$R_s = 1000\left[V_s(C_n - C_0) + \sum_{i=1}^{n} V_{i-1}(C_{i-1} - C_a)\right]/W_s \tag{2}$$

where $R_s$ is the amount of TOC, TN, TP and $NH_3$-N (mg/kg) released from pond sediment under static conditions; vs. is the volume (800 mL) of the water sample in the release experiment under static conditions; $V_{i-1}$ denotes the volume (150 mL) of the water sample

taken at $i$-1 time; $C_0$ denotes the initial concentrations in the solution; $C_n$ and $C_{i-1}$ denote the nutrition concentration (mg/L) of the overlying water sample taken at $n$ and $i$-1 time, respectively; $C_a$ is the nutrition concentration (mg/L) in the supplement water; and $W_s$ is the quality of the dried sediment (mg) in the release experiment under static conditions.

### 2.3. Analytical Methods

The contents of total nitrogen (TN), total phosphorus (TP) and TOC in sediment samples were determined according to national or professional standards, that is Soil quality-Determination of total nitrogen-Modified Kjeldahl method (HJ 717-2014), Soil-Determination of Total Phosphorus by alkali fusion -Mo-Sb Anti-spectrophotometric method (HJ 632-2011) and Soil testing. Part 6: Method for determination of soil organic matter (NY/T 1121.6-2006), respectively. The concentrations of TOC, DTN, DTP and $NH_3$-N in water samples were determined, referring to Water quality-Determination of total organic carbon-Combustion nondispersive infrared absorption method (HJ 501-2009), Water quality-Determination of total nitrogen-Alkaline potassium persulfate digestion UV spectrophotometric method (HJ 636-2012), Water quality-Determination of total phosphorus-Ammonium molybdate spectrophotometric method (GB/T 11893-1989) and Water quality-Determination of ammonia nitrogen—Nessler's reagent spectrophotometry (HJ 535-2009), respectively. Blank samples and quality control samples were conducted for each batch of water samples and sediment samples.

### 2.4. Data Analysis

The graphics and regression of data were generated with Origin 2018 (Origin Lab, Northampton, MA, USA). Pearson correlation analysis was carried out by using SPSS 26.0 (IBM, Chicago, IL, USA) to identify the relationships between the maximum release amounts and original concentrations in sediment samples.

## 3. Results and Discussion

### 3.1. Contents of TN, TP and TOC in Sediment Samples

The vertical distribution characteristics of nutrients and TOC in sediments of fish pond and lotus pond were obviously different. Shown in Figure 1a, the contents of total nitrogen in surface sediment (0–10 cm) of the fish pond and the lotus pond were the highest, reaching 1810 and 1760 mg/kg, respectively. The total nitrogen content of the fish pond decreased gradually from surface to bottom, but was significantly higher than that of the lotus pond. The concentration of the middle layer of the fish pond was also high, reaching 1560 mg/kg. The TN content in the middle and bottom layers of the lotus pond was similar. As shown in Figure 1b, the contents of TP in the surface sediment of the fish pond and lotus pond were the highest, 1463 and 1370 mg/kg, respectively. The contents of total phosphorus in the fish pond and lotus pond decreased from the surface to the bottom, which was similar to the varying of total nitrogen. Figure 1c shows the variation of TOC contents in the columnar sediments. The content in the surface layer of the fish pond was as high as 21.2 g/kg, and decreased rapidly from the surface layer to the bottom layer. The TOC content in the surface layer of the lotus pond sediment was relatively high, 10.1 g/kg, and the contents in the middle and bottom layer were basically equivalent to those in the surface layer. In general, the contents of TP and superficial TN in fish pond samples were like those in the lotus pond ($p > 0.05$), while the contents of TN in the middle and bottom layers were higher than the values in the lotus pond. The TOC content in the surface layer of the fish pond was significantly higher than the lotus pond. TP and TOC in the bottom layer of the lotus pond were significantly higher than the fish pond.

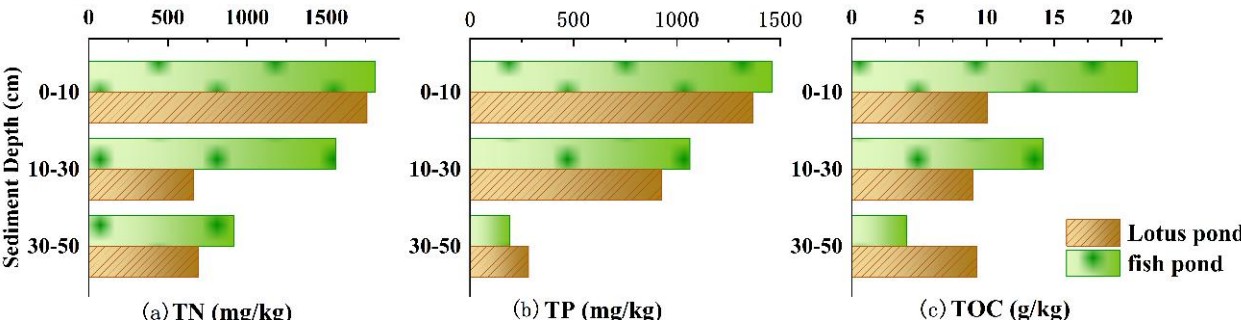

**Figure 1.** Nutrient contents in sediment samples of fish pond and lotus pond; (**a**) TN contents; (**b**) TP contents; (**c**) TOC contents.

There were few studies on the contents of TN, TP and TOC of different depths in pond sediment previously, and most of them were focused on the surface layer [33]. In this study, the highest TN contents in both sediments of the fish pond and the lotus pond were concentrated in surface samples, which were similar to the median level of TN contents in the study of freshwater aquaculture ponds in Shanghai from 2016 to 2019 (287.24–2683.24 mg/kg) [31]. TN, TOC and TP contents in surface sediments were all higher than those in sea cucumber and jellyfish pond sediments in Ren's study [34]. Meanwhile, TN and TP contents were higher than the values of crab, clam, fodder and shrimp pond sediments in Wu's study [35]. In this study, TN, TP and TOC of surface sediments of fish pond were significantly higher than those of middle and deep sediments, which was consistent with previous studies [36,37].

Compared with the surrounding area located in Taihu basin, the contents of TN, TP and TOC in the surface sediments of the fish pond were 1.63, 1.57 and 2.37 times higher than the average values in Zhushan Bay, respectively. The TN, TP and TOC contents of surface sediment samples in the lotus pond were 1.59, 1.47 and 1.13 times the average content of Zhushan Bay, respectively [38]. The surface sediment samples of the fish and lotus pond were also higher than Lake Taihu when compared with previous studies [24,39]. Therefore, it is necessary to strengthen dredging and management of nutrients and organic matter in the process of returning polder to lake.

### 3.2. Release under the Disturbance Conditions

The release amounts of TOC, DTP, $NH_3$-N and DTN in sediment samples under disturbance are shown in Figure 2. The release of TOC, DTP, $NH_3$-N and DTN in the surface layer of the lotus pond was the highest, followed by the bottom sediments, and the release of the middle layer was the lowest. The release of TN and TOC in different layers from the fish pond was similar, and was lower than the lotus pond. In the release experiment under disturbance conditions, the surface sediments of the lotus pond had the highest release amounts of TOC, DTP, $NH_3$-N and DTN. The release of nutrients (DTP, $NH_3$-N and DTN) in the bottom sediment of the lotus pond was higher than that of the middle layer, while the release of TOC in the middle sediment exceeded the bottom sediment. There was a continuous release of TOC, DTP, $NH_3$-N and TN in the sediment of the lotus pond, while the release of nutrients was only rapid at its initial stage in the fish pond. It is worth mentioning that the amounts of TOC and nutrients released from the sediments in the lotus pond were significantly higher than those in the fish pond, despite their relatively low amount.

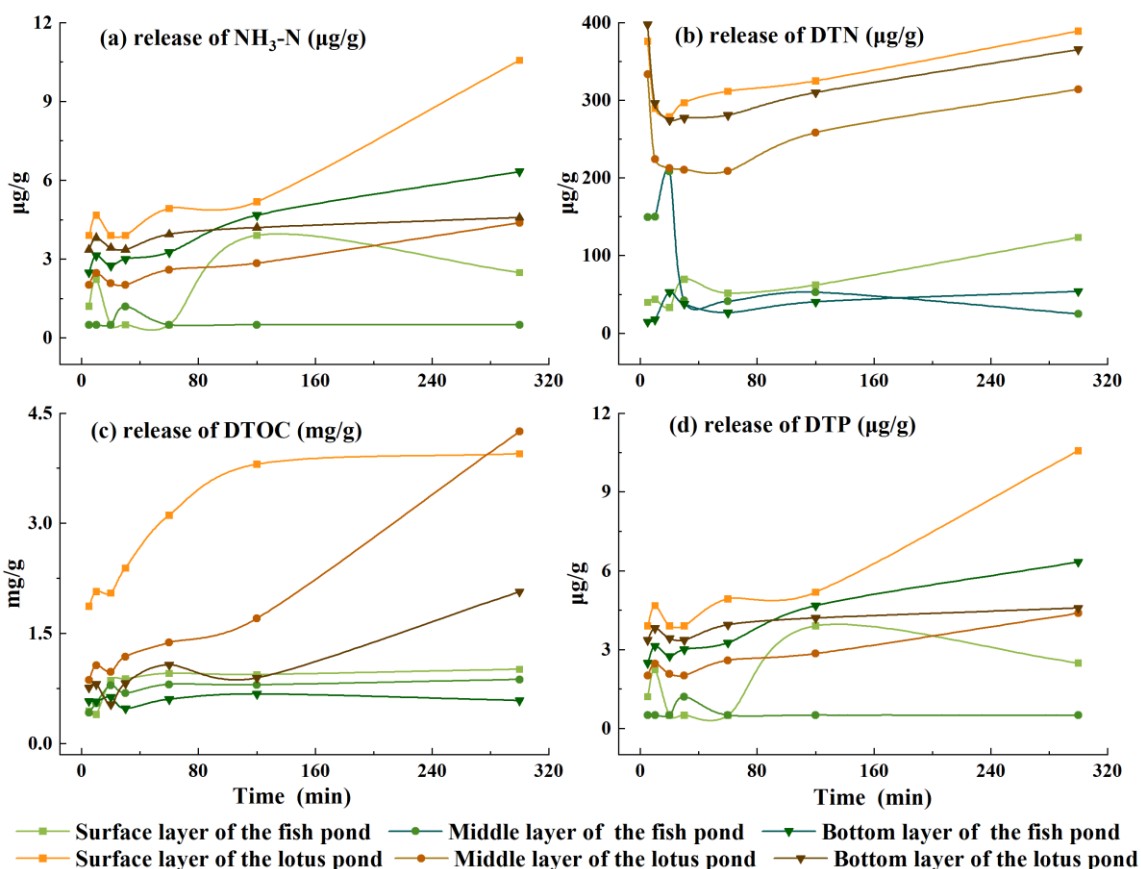

**Figure 2.** Release of NH$_3$-N, DTN, DTOC and DTP under disturbance; (**a**) NH$_3$-N; (**b**) DTN; (**c**) DTOC; (**d**) DTP.

Table 1 lists the maximum release rates and the corresponding time periods under the disturbed condition. Except DTN released from the bottom sediment, the maximum release rates basically occurred in the initial period (0~5 min). After the rapid release of NH$_3$-N in the early stage, the subsequent release rate slowed down, which was similar to previous studies [40], which could be explained by release reaction composed of fast reaction and slow reaction. Furthermore, the rates of the lotus pond sediment were overall higher than the fish pond. Combined with the higher amount of disturbed release, this may indicate that the sediment from the lotus pond has greater release potential risk of nutrients and carbon than the fish pond.

**Table 1.** The maximum release rates and the corresponding time periods under the disturbed conditions.

| Maximum Release Rate (µg/(g·min)) | 1# Fish | 2# Fish | 3# Fish | 1# Lotus | 2# Lotus | 3# Lotus |
|---|---|---|---|---|---|---|
| NH$_3$-N | 2.75 | 2.75 | 2.37 | 20.48 | 9.77 | 2.42 |
| Time periods | 0–5 min | 0–5 min | 0–5 min | 0–5 min | 0–5 min | 0–5 min |
| DTN | 7.91 | 29.89 | 3.56 | 75.15 | 66.62 | 79.47 |
| Time periods | 0–5 min | 0–5 min | 20–30 min | 0–5 min | 0–5 min | 0–5 min |
| DTP | 0.24 | 0.10 | 0.50 | 0.78 | 0.40 | 0.67 |
| Time periods | 0–5 min | 0–5 min | 0–5 min | 0–5 min | 0–5 min | 0–5 min |
| TOC | 88.8 | 84.2 | 115.4 | 374 | 173.4 | 152.4 |
| Time periods | 0–5 min | 0–5 min | 0–5 min | 0–5 min | 0–5 min | 0–5 min |

1#: the surface layer, 2#: the middle layer, 3#: the bottom layer.

### 3.3. Release under Static Conditions

The release amounts of TOC, DTP, NH$_3$-N and DTN in sediment samples under static conditions are shown in Figure 3. Unlike the results of the disturbed experiment, nutrients (DTP, NH$_3$-N and DTN) in the surface sediment from the fish pond were higher than the bottom and middle layers. The release of DTP, NH$_3$-N and DTN from the surface layer of the lotus pond was similar, and was significantly higher than that of the middle and bottom layers. At the end of the experiment, both fish pond and lotus pond received the highest release amount of NH$_3$-N and DTN. In general, the release amount under static conditions of sediment in the lotus pond was higher than that in the fish pond, which was consistent with the results under disturbance conditions.

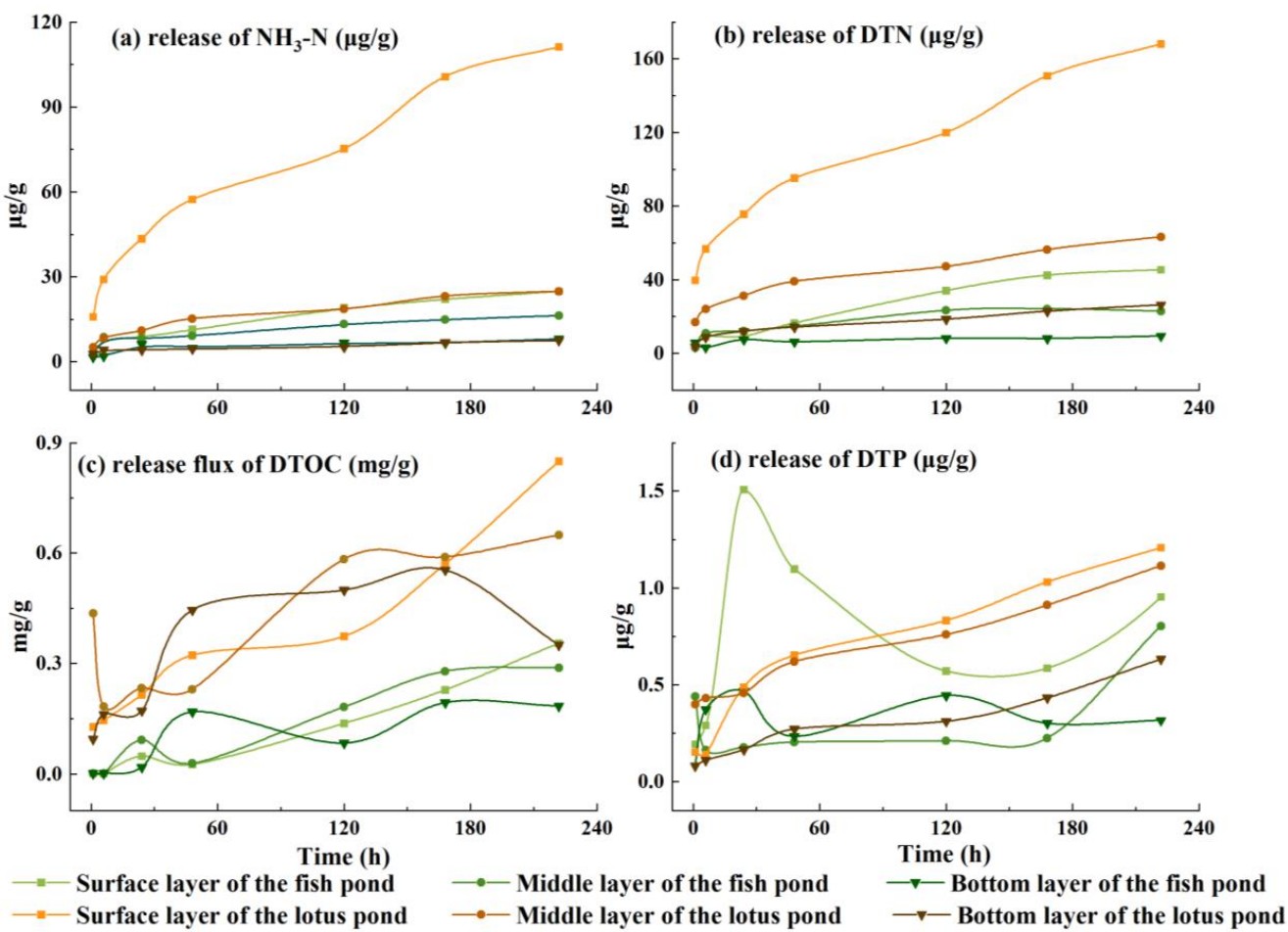

**Figure 3.** Release of NH$_3$-N, DTN, TOC and DTP under static condition; (**a**) NH$_3$-N; (**b**) DTN; (**c**) TOC; (**d**) DTP.

Nutrient release of the fish pond and the lotus pond increased steadily (except for DTP released from the surface sediments of the fish pond) during the whole period. DTP in the surface and bottom sediments of the fish pond increased first within 24 h, then decreased and finally stabilized, while the release of DTP in the lotus pond gradually increased. TOC release in the sediments of the two types of ponds fluctuated and increased with time. The sediment of the lotus pond had higher release potential than the fish pond, especially for the NH$_3$-N and DTN released by the surface sediment of the lotus pond.

Table 2 lists the maximum release rates and the corresponding time periods in the static release experiment. The maximum release rates of the lotus pond were generally higher than those of the fish pond, and occurred at the initial stage. In addition, the maximum

release rates of $NH_3$-N, DTN and TOC of the lotus pond were significantly higher than the fish pond, which was also consistent with the results under the disturbed condition.

**Table 2.** Maximum release rates in static release experiment.

| Maximum Release Rate (μg/(g·h)) | 1# Fish | 2# Fish | 3# Fish | 1# Lotus | 2# Lotus | 3# Lotus |
|---|---|---|---|---|---|---|
| $NH_3$-N | 3.54 | 2.63 | 1.55 | 15.86 | 5.14 | 2.85 |
| Time periods | 0–1 h | 0–1 h | 0–1 h | 0–1 h | 0–1 h | 0–1 h |
| DTN | 3.94 | 3.03 | 5.52 | 39.64 | 16.88 | 3.49 |
| Time periods | 0–1 h | 0–1 h | 0–1 h | 0–1 h | 0–1 h | 0–1 h |
| DTP | 0.19 | 0.44 | 0.08 | 0.15 | 0.40 | 0.08 |
| Time periods | 0–1 h | 0–1 h | 0–1 h | 0–1 h | 0–1 h | 0–1 h |
| TOC | 2.61 | 5.09 | 6.27 | 127.84 | 436.32 | 93.92 |
| Time periods | 6–24 h | 6–24 h | 24–48 h | 0–1 h | 0–1 h | 0–1 h |

1#: the surface layer, 2#: the middle layer, 3#: the bottom layer.

In this study, the release under disturbed conditions was stronger than the static release, which was likewise revealed in previous studies [41]. In actual shallow lakes, the release mode of nutrients in sediments is largely controlled by hydrodynamics to a great extent, and is also affected by various environmental factors. Due to adsorption and flocculation, part of the nutrients would return to the bottom of the water body along with the precipitation of suspended matter. Under reduction conditions, organic particles are degraded and separated. Sediments will wait for the subsequent wave and wind to force them to resuspend and thus release nutrients [41].

*3.4. Characteristics Analysis of the Release of the Pond Sediment*

Ammonium release from the sediments has significant effects on water quality in eutrophic lakes [40]. Furthermore, ammonium is the dominant form of inorganic nitrogen which releases from sediments [42,43]. In this study, the release of $NH_3$-N variations with time can be expressed as regression equations (Table 3). Most of the regression equations had a coefficient of determination ($R^2$) more than 0.8, implying a reasonable fitting effect.

**Table 3.** The regression equations for the release flux of $NH_3$-N.

| Type of Pond | Condition | Sediment Depth (cm) | Regression Equation | $R^2$ |
|---|---|---|---|---|
| fish | disturbance | 0–10 cm | $q = 22.61 - \exp(2.43 - 0.055t)$ | 0.91 |
| fish | disturbance | 10–30 cm | $q = 21.96 - \exp(2.14 - 0.016t)$ | 0.94 |
| fish | disturbance | 30–50 cm | $q = 58.24 - \exp(3.82 - 0.006t)$ | 0.97 |
| lotus | disturbance | 0–10 cm | $q = 148.42 - \exp(4.08 - 0.062t)$ | 0.86 |
| lotus | disturbance | 10–30 cm | $q = 81.57 - \exp(3.60 - 0.008t)$ | 0.83 |
| lotus | disturbance | 30–50 cm | $q = 30 - \exp(2.82 - 0.002t)$ | 0.50 |
| fish | static | 0–10 cm | $q = 31.83 - \exp(3.29 - 0.006t)$ | 0.98 |
| fish | static | 10–30 cm | $q = 17.66 - \exp(2.53 - 0.009t)$ | 0.88 |
| fish | static | 30–50 cm | $q = 6.93 - \exp(1.76 - 0.044t)$ | 0.86 |
| lotus | static | 0–10 cm | $q = 137.17 - \exp(4.75 - 0.006t)$ | 0.98 |
| lotus | static | 10–30 cm | $q = 26.47 - \exp(3.00 - 0.01t)$ | 0.98 |
| lotus | static | 30–50 cm | $q = 20 - \exp(2.81 - 0.001t)$ | 0.93 |

The above regression equations can be expressed in the following form:

$$q = c - e^{(a-bt)} \tag{3}$$

where $q$ is the release flux, and $a$, $b$ and $c$ are regression coefficients. Due to the different experimental durations, the unit of time ($t$) in the static fitting regression is minute; the

unit of time (*t*) in the disturbance fitting is hour. It could be indicated that the value of coefficient *c* in each equation is expressed as the maximum value of each release model, that is, the maximum amount released in the corresponding release experiment. In the aim to further investigate the characteristics of the release from the pond sediments, the correlations between maximum release amounts and original concentrations in sediment samples are analyzed and listed in Table 4.

**Table 4.** Correlation matrix between the maximum release amounts and other factors.

|  | DT | JT | DTN | DTP | TOC | Depth |
|---|---|---|---|---|---|---|
| DT | 1 |  |  |  |  |  |
| JT | 0.852 * | 1 |  |  |  |  |
| DTN | 0.128 | 0.526 | 1 |  |  |  |
| DTP | 0.261 | 0.560 | 0.811 * | 1 |  |  |
| TOC | −0.395 | 0.034 | 0.685 | 0.754 * | 1 |  |
| Depth | −0.333 | −0.569 | −0.751 * | −0.991 ** | −0.679 | 1 |

* The correlation was significant at 0.05 level. ** The correlation was significant at 0.01 level.

The maximum release amounts in sediments are not significantly correlated with sediment concentrations, while the maximum release amounts under disturbance were significantly correlated with the amount of static state ($p < 0.05$), indicating that the two release modes are to some extent comparable. In addition, the content of TP in sediments was significantly positively correlated with the contents of TN and TOC ($p < 0.05$), and the contents of TP and TN were significantly negatively correlated with sampling depth ($p < 0.01$ and $p < 0.05$, respectively).

## 4. Conclusions

Agricultural/aquacultural activities have led to much higher TN, TP and TOC in sediments from the fish pond and the lotus pond in Zhushanwei area than the values from the adjacent aquatic area. The vertical distribution of TN, TP and TOC in the sediments sampled from the two types of ponds is obviously different. The N, P and organic matter in the fish pond were mainly concentrated in the surface and middle layers of the sediments. The values of TP and TN in the surface layer of the two types of ponds were similar, while TN and TOC in the middle layer and bottom layer of the fish pond were significantly higher. TOC in the bottom sediment of the lotus pond was generally the same as that in the surface layer. Under disturbed conditions, the release of NH$_3$-N, DTN and TOC was higher than that under static conditions. Sediments of the lotus pond had higher release rates than the fish pond under disturbed conditions. The amount of nutrients released from lotus pond sediment was higher than that of the fish pond. The release in the surface layer of the lotus pond was the highest, while the amount released from the other layers was much lower. Under static conditions, NH$_3$-N and DTN in surface sediments of two ponds released more than those from the bottom layers, especially in the lotus pond. DTP in the surface and bottom sediments of the fish pond increased rapidly within 24 h, while the release of DTP in the lotus pond represented a gradual increase throughout the experimental period. The maximum release under both disturbance and static conditions was significantly correlated ($p < 0.05$). Despite the lower concentration of nutrients and TOC, the lotus pond presented higher release. Due to the obvious differences in the content, vertical distribution and release characteristics of the fish pond and lotus pond, different management strategies should be used for future management practice and environmental risk evaluation during the project of returning polder to lake.

**Author Contributions:** Conceptualization, C.P., Y.G., F.L. and D.Z.; software, C.P.; validation, C.P. and Y.G; formal analysis, C.P.; investigation, C.P., J.H. and Y.Y.; resources, Y.T., G.S. and D.S.; data curation, C.P.; writing—original draft preparation, C.P; writing—review and editing, F.L. and Y.G.; visualization, C.P.; supervision, D.Z. and H.T.; project administration, H.T., Y.T. and G.S.; funding acquisition, Y.T., G.S. and D.S. All authors have read and agreed to the published version of the manuscript.

**Funding:** The study was funded by the Research Project of China Three Gorges Corporation (Research on Key Technology of Urban Water Environment Monitoring System and Pollution Traceability), grant number (202003134).

**Institutional Review Board Statement:** Not applicable.

**Informed Consent Statement:** Not applicable.

**Data Availability Statement:** All data used in this study have been presented in the manuscript.

**Conflicts of Interest:** The authors declare no conflict of interest.

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
