# Peer review of "Pollution and Release Characteristics of Nitrogen, Phosphorus and Organic Carbon in Pond Sediments in a Typical Polder Area of the Lake Taihu Basin"

_water, doi:10.3390/w14050820_

Round 1
Reviewer 1 Report
The topic of this manuscript is interesting and falls within the aim and scope of the journal. The abstract should be rewritten in a way to attract the readers. Some sentence are too long and should be revised. My main doubt is the novelty of this research because there are already many publications regarding this issue. Authors must clearly justify this and highlight it in the final part of the introduction, together with the goals of the study. Authors should improve the quality of Introduction section by inserting more references. This will give more international flavour in the sections of Introduction and Discussion.
Additional comments are included in the attached .pdf file.
Regards,

Author Response
感谢您的评论。请参阅附件。

Reviewer 2 Report
Line 2: Please revise the Title. The first word should be “Pollution”.
Line 33: Usually, the keywords should be different from the words used in the paper’s title. Please revise.
Line 38-41: I couldn’t understand this statement (… one of the important for?). Please revise.
Line 44: Eutrophication and algal blooms are consequences of nutrients. Please revise.
Line 55: sediments nutrients? Please revise.
Line 59: Nutrients characteristics? Please revise.
Line 66: Please revise.
Line 74: What do you mean by surface sediments? Suspended sediments?
Line 88: Please remove “in this study”.
Section 2.4.: Which statistical methods you used? This section should be rewritten in a clear way.
Tables 1 and 2: Please revise the unit given for “maximum release rate”.
The authors should place this work into a broader setting. The literature review, discussion, and justifications made through the manuscript does help in this regard. I suggest the authors to support their study with referring to “Water 2021, 13(20), 2874”.
Author Response
Thanks for your comment. Please see the attachment.

Round 2
Reviewer 1 Report
The manuscript has reached the level of acceptance for publication in Water.
Reviewer 2 Report
All of my comments have been responded properly.
My suggestion is acceptance.
Good luck.